# Mesotrode allows chronic simultaneous mesoscale cortical imaging and subcortical or peripheral nerve spiking activity recording in mice

**Dongsheng Xiao[1,2], Yuhao Yan[1,2], Timothy H Murphy[1,2]\***

[1]University of British Columbia, Department of Psychiatry, Kinsmen Laboratory of Neurological Research, Vancouver, Canada; [2]Djavad Mowafaghian Centre for Brain Health, University of British Columbia, Vancouver, Canada

**\*For correspondence:** thmurphy@mail.ubc.ca

**Competing interest:** The authors declare that no competing interests exist.

**Preprint posted** 03 February 2023
**Sent for Review** 01 April 2023
**Reviewed preprint posted** 31 May 2023
**Version of Record published** 14 November 2023

**Abstract** Brain function originates from hierarchical spatial-temporal neural dynamics distributed across cortical and subcortical networks. However, techniques available to assess large-scale brain network activity with single-neuron resolution in behaving animals remain limited. Here, we present Mesotrode that integrates chronic wide-field mesoscale cortical imaging and compact multi-site cortical/subcortical cellular electrophysiology in head-fixed mice that undergo self-initiated running or orofacial movements. Specifically, we harnessed the flexibility of chronic multi-site tetrode recordings to monitor single-neuron activity in multiple subcortical structures while simultaneously imaging the mesoscale activity of the entire dorsal cortex. A mesoscale spike-triggered averaging procedure allowed the identification of cortical activity motifs preferentially associated with single-neuron spiking. Using this approach, we were able to characterize chronic single-neuron-related functional connectivity maps for up to 60 days post-implantation. Neurons recorded from distinct subcortical structures display diverse but segregated cortical maps, suggesting that neurons of different origins participate in distinct cortico-subcortical pathways. We extended the capability of Mesotrode by implanting the micro-electrode at the facial motor nerve and found that facial nerve spiking is functionally associated with the PTA, RSP, and M2 network, and optogenetic inhibition of the PTA area significantly reduced the facial movement of the mice. These findings demonstrate that Mesotrode can be used to sample different combinations of cortico-subcortical networks over prolonged periods, generating multimodal and multi-scale network activity from a single implant, offering new insights into the neural mechanisms underlying specific behaviors.

## eLife assessment

This **valuable** study combines chronic widefield calcium imaging of dorsal cortex activity at the mesoscale level with electrical recording of single neurons in specific cortical and subcortical locations. This work provides **compelling** evidence for recording neuronal activity at multiple temporal and spatial scales by combination of optical and electrophysiological methods. This work will be of broad interest to system neuroscientists studying neural circuits.

## Introduction

System-level mechanisms of cognition and action across networks of single neurons in awake, behaving mice remain largely elusive (*Buzsáki, 2006*; *Aru et al., 2020*; *Roth and Ding, 2020*). The difficulty comes from recording neural activity over large spatial scales but with single neuron resolution during

sensory and motor processes (*Alivisatos et al., 2012*). In the mammalian brain, large-scale cortical network activity is dynamically sculpted by local or long-range inputs from individual neurons in various cortical or subcortical structures (*Koch et al., 2016*). Decoding the principles of neuronal network activity is essential for understanding brain function (*Oh et al., 2014*; *Jiang et al., 2015*). One important aspect of such understanding is mapping the functional connectivity of single neurons in relation to cortical networks. Emerging studies using in vivo electrophysiological and imaging techniques have revealed that activities of single neurons are functionally coupled to those of local microcircuits or the global cortical networks, and such connectivity is dynamically regulated depending on the behavioral state of the animal (*Barson et al., 2018*; *Chen et al., 2013a*; *Xiao et al., 2017*; *Clancy et al., 2019*). While most of the current studies only provide a snapshot of the connectivity map of a single neuron, the development of procedures where both single-unit activity and large-scale cortical network dynamics of awake-behaving mice can be chronically recorded is desirable.

Previously, chronic extracellular recordings in rodent brains were achieved most widely using bundled microwires called tetrodes, and more recently extended to the use of high-throughput recording devices such as silicon probes including Neuropixels (*Bragin et al., 2000*; *Nguyen et al., 2009*; *Vandecasteele et al., 2012*; *Voigts et al., 2013*; *Delcasso et al., 2018*; *Juavinett et al., 2019*; *Steinmetz et al., 2021*). In the case of conventional chronic tetrode recording setups, a common feature is the inclusion of a microdrive that provides axial control over the positioning of the electrodes which allows the electrode position to be adjusted for better recording qualities throughout the experimental periods. However, this microdrive hinders simultaneous wide-field optical imaging of the cortex, making it difficult to investigate the correlation between subcortical single-unit activation and cortical network activity. On the other hand, chronic silicon probe recordings rely on a skull-mounted probe and a large headstage, also have the same limitation of not allowing for simultaneous wide-field imaging.

To overcome these challenges, we designed the Mesotrode system where we combine in vivo electrophysiology via multi-site tetrode implants and mesoscale brain imaging to characterize the functional connectivity of individual cortical or subcortical neurons chronically. We show that these tetrode preparations can obtain high-fidelity, single-unit activity in any cortical or subcortical structure. More importantly, this setup preserves optical access to the entire cranial window of the animal which permits simultaneous functional brain imaging as well as optogenetic manipulation of neuronal activity across the whole dorsal cortex. To obtain high-resolution wide-field imaging of mouse brain activity, we chose to utilize GCaMP6, a genetically encoded $Ca^{2+}$ indicator (GECI) widely used to optically record suprathreshold neuronal activation due to its high sensitivity and optimal signal-to-noise ratio (*Chen et al., 2013b*). Furthermore, it has been shown that GECIs can stably report neuronal activity over several months (*Huber et al., 2012*; *Margolis et al., 2012*; *Silasi et al., 2016*), making them ideal for chronic studies of cortical network dynamics. We found that neurons from various subcortical structures, including the hippocampus, thalamus, striatum, and other midbrain areas, display distinct functional connectivity patterns with the cortex. More importantly, we show that the spike-triggered maps (STMs) of recorded neurons can be stable for up to two months. Moreover, we extended Mesotrode to record facial nerve spiking activity and identified a novel cortical network that is causally involved in controlling facial movement, further highlighting the wide applicability of this technique. These results indicate that our Mesotrode can be widely exploited to investigate multiscale functional connectivity within the central nervous system of mice over an extended timescale.

## Results

### Chronic single-unit recording with wide-field transcranial imaging window

We developed the Mesotrode implantation procedure with the aim to make them low profile, flexible in location, minimally invasive, and yet stable while maintaining optical access to the entire dorsal cortex such that we can record chronic single neuron activity and mesoscale cortical dynamics simultaneously. Briefly, we fabricated tetrodes from tungsten wires which were then soldered to a miniature connector that was cemented onto the back of the mouse skull during surgery (see Materials and methods). The single tetrode was stiff enough to be straightly inserted into the deep brain areas through a small burr hole (<1 mm; *Figure 1A*). The body of the tetrode that remained outside of the

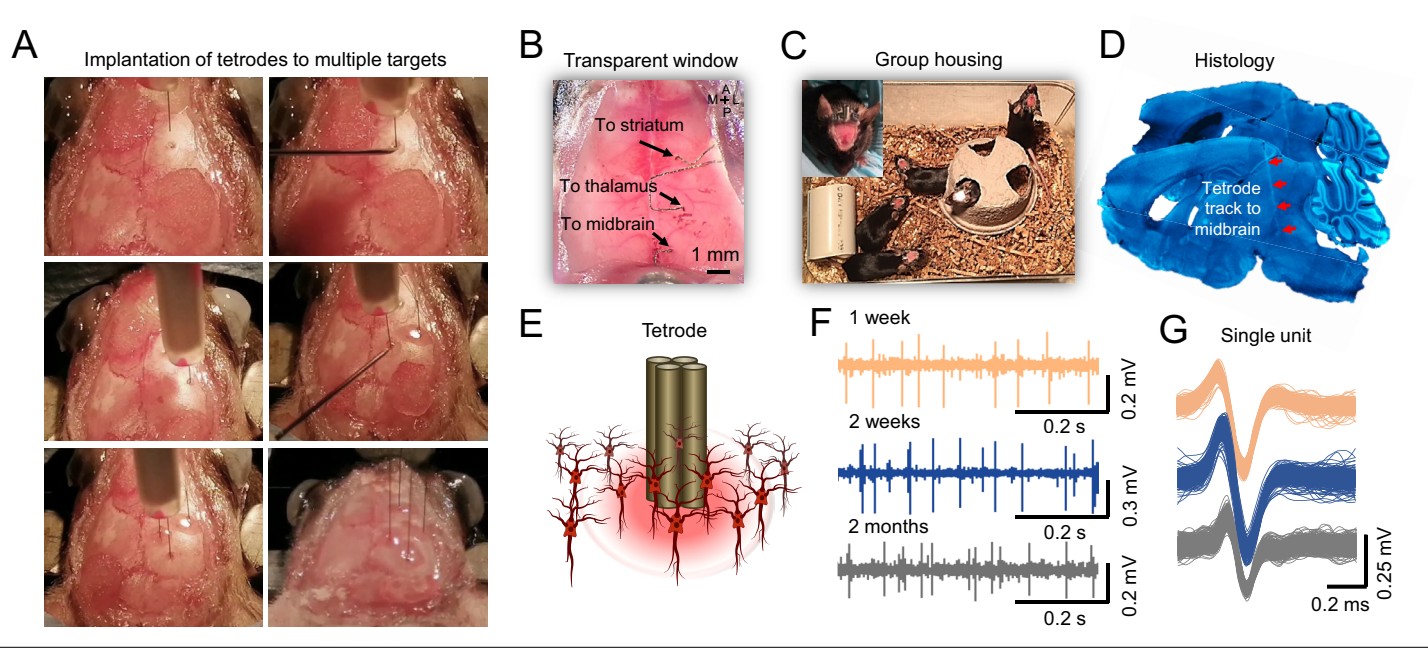

**Figure 1.** Chronic tetrodes recording compatible with mesoscale transcranial Imaging. (**A**) Images of skull surface during tetrode implantation. Panels depict the location of 4 tetrodes placed in skull burr holes using a micromanipulator. (**B**) Image showing transcranial window with tetrodes placed in the striatum, thalamus, and midbrain. (**C**) Chronic tetrode recording is compatible with group-housed mice. (**D**) Post-mortem histology is performed to confirm the tetrode location. (**E**) Cartoon of tetrode and local neurons. (**F**) Example of long-term recording using transcranial imaging window plus tetrodes. (**G**) Sorted spikes from (**F**) (brown, blue, and gray indicate 1 week, 2 weeks, and 2 months after tetrode implantation).

brain was bent parallel to the skull and the minimal size helped preserve optical transparency during imaging sessions (*Figure 1B*). As the miniature connectors are low-profile, mice were group-housed after the surgery (*Figure 1C*), and we did not observe significant damage to the window or the connectors after long-term monitoring. The tetrodes can be flexibly implanted in almost any brain structure of interest, either cortical or subcortical. We have successfully obtained high-quality recordings in the mouse cortex, hippocampus, thalamus, striatum, and midbrain. The recording site of each tetrode is confirmed using *post-hoc* histological analysis (*Figure 1D*). We found that the single unit activity was relatively stable during 1 week, 2 weeks, and 2 months of recordings after implantation (*Figure 1F and G*), which is consistent with previously reported chronic tetrode recordings in mice (*Tolias et al., 2007*; *Hong and Lieber, 2019*; *Kim et al., 2020*; *Voigts et al., 2020b*).

We implanted a total of 29 tetrodes in the midbrain, hippocampus, thalamus, striatum, and cortex of 14 mice, and recorded the activity of 110 neurons with simultaneous mesoscale cortical imaging for up to 2 months after implantation (*Figure 2*). We registered all the putative positions of the recorded neurons as well as the tetrode tracks to a 3D mouse brain model, which was reconstructed from an MRI scan (*Egan et al., 2015*; *Figure 2A and B*). First, we compared the firing profiles of the neurons from different brain regions. On average, we obtained 3–5 neurons per tetrode implanted, and this yield was consistent across regions (*Figure 2C*). The inter-spike interval (ISI) of neurons recorded from most brain regions ranged between 0.2 and 0.7 s, except for neurons of the striatum, which fires significantly less compared to neurons in other brain regions (*Figure 2D*, $p < 0.05$). The coefficient of variation (CV) of ISI, a metric that indicates the consistency of the firing rate, was similar across brain regions (*Figure 2E*).

## Spike-triggered average mapping of the thalamus, striatum, hippocampus, and midbrain neuron-defined mesoscale cortical networks

Spike-triggered average mapping (STM) has been previously used to investigate functional connectivity between single spiking neurons and cortical networks (*Barson et al., 2018*; *Xiao et al., 2017*; *Clancy et al., 2019*; *Liu et al., 2021a*). However, this has only been achieved using acute preparation.

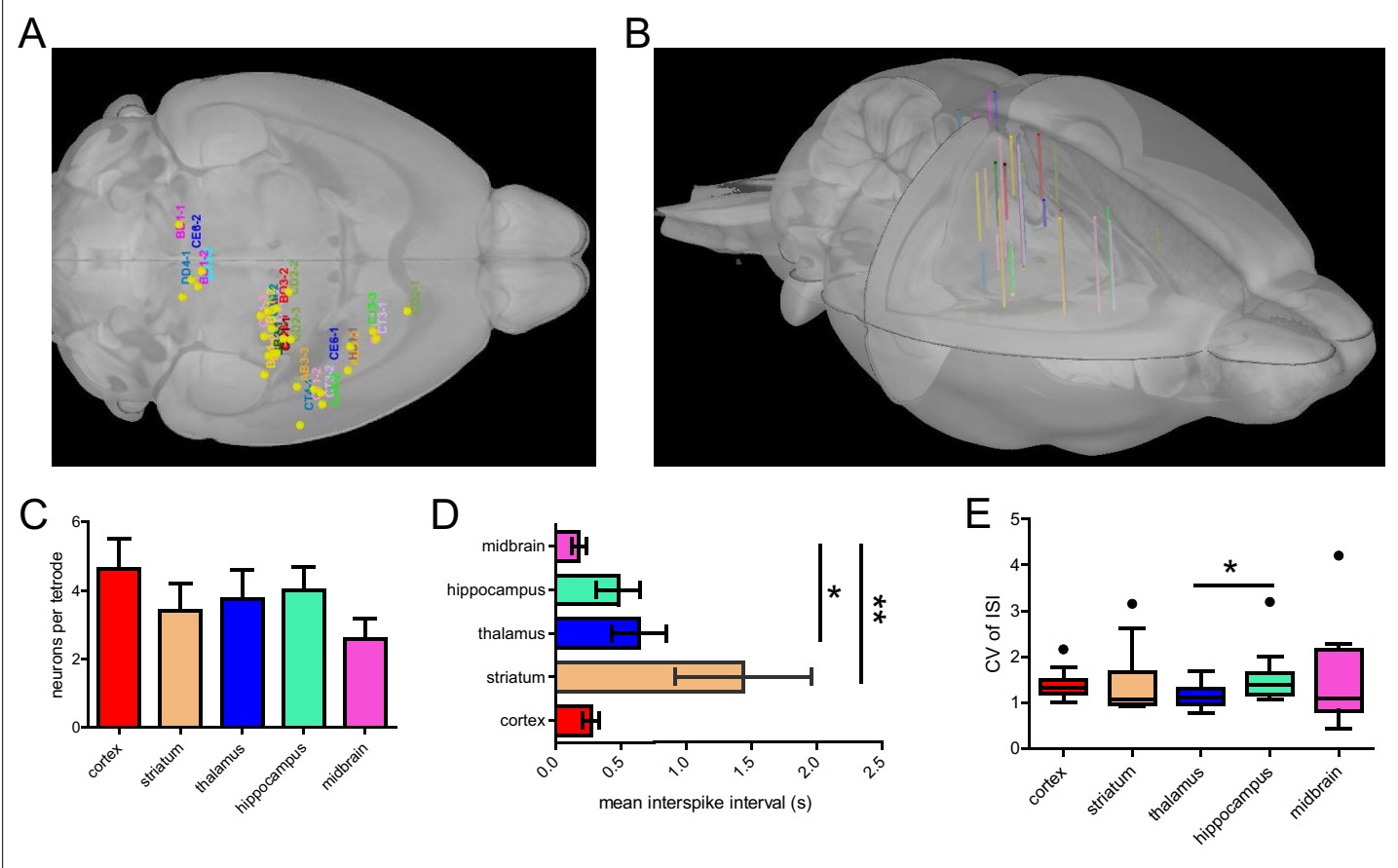

**Figure 2.** Group statistics on tetrode implantation across mice. (**A**) Labels corresponding to mouse IDs (n=14 mice) were registered to a 3D mouse brain model based on tetrode locations. (**B**) Side view of the 3D reconstruction of 29 tetrode tracks implanted in 14 mice registered to the 3D mouse brain model. (**C**) The mean number of well-isolated neurons recorded per tetrode implanted in different brain regions. (**D**) Mean inter-spike interval (ISI) of neurons recorded in different brain regions (Dunn's Multiple Comparison Test, striatum vs midbrain, p<0.01, thalamus vs midbrain, p<0.05, n=14 mice). (**E**) Mean coefficient of variation of ISIs of neurons across brain regions (Dunn's Multiple Comparison Test, thalamus vs hippocampus, p<0.05, n=14 mice).

To assess the functional connectivity of the chronically recorded neurons, we combined electrophysiological recordings with simultaneous mesoscale cortical imaging through a bilateral window that encompassed the entire dorsal cortex in head-fixed, awake mice (*Figure 3A and B*, *Videos 1 and 2*). To obtain STMs of the recorded neuron, we calculated the peak response of normalized Ca²⁺ activity ($\Delta F/F$) of each pixel averaged between 3 s before and after (±3 s) the onset of each spike, which gave us a wide-field mapping of the cortical areas that were associated with spiking activity of the recorded neuron (*Xiao et al., 2017*). Green reflectance signals, which were recorded with the same frequency as the GCaMP6 signal, were used to correct hemodynamic artifacts (*Figure 3C and D*).

We found that single neuron defined functional maps were stable across recording sessions and days. For example, the STM of a hippocampus neuron was associated with the RSP, BCS, and M2 region for 10 recording sessions on different days (*Figure 4A and B*). To better visualize the stability of STMs across recording, we used Mesonet, a machine-learning based toolbox for parcellating brain regions, to accurately align our mesoscale images to a common brain atlas (*Xiao et al., 2021*). We show that the STMs of a midbrain neuron were relatively stable over 2 months (*Figure 4C and D*). The spiking activity of the neuron on different days was highly correlated with the lower limb, upper limb, and trunk sensorimotor areas on both hemispheres of the cortex. To examine the neural correlates of different behaviors, we also incorporated a Raspberry pi camera to monitor the spontaneous behavior of the mice during brain activity recordings (*Figure 4C and E*). In this example, we show that the instantaneous firing rate and patterns of this midbrain neuron were highly correlated with the paw movement of the mouse, consistent with our finding that this neuron is functionally associated with

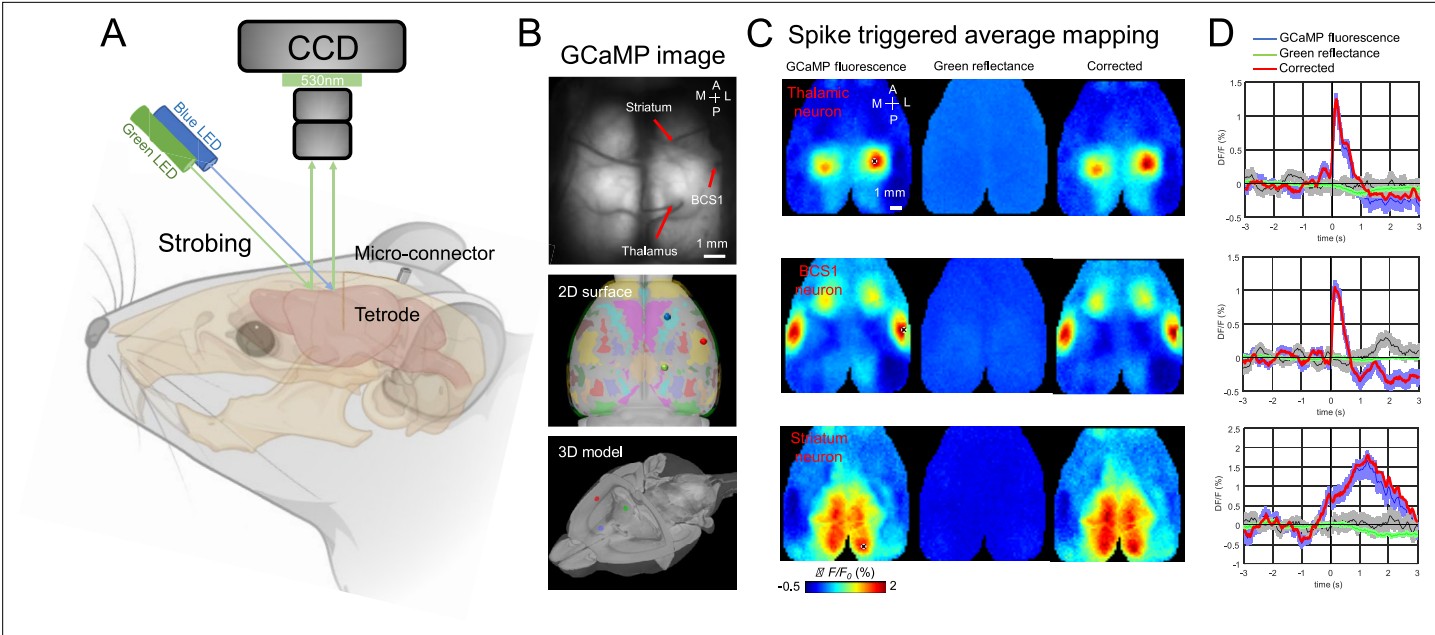

**Figure 3.** Recording setup and example spike-triggered average maps. (**A**) Illustration of experiment setup incorporating wide-field imaging, and tetrode recording with simultaneous behavioral monitoring. (**B**) Example tetrodes implantation in the thalamus, BCS1, and striatum. The middle and bottom panel shows the tetrode location registered in a 3D model of the mouse brain. (**C**) Spike-triggered average of GCaMP6s fluorescence (left), hemodynamic signal (middle, green reflectance), and hemodynamic-subtracted cortical maps (right) of thalamic, BCS1, and striatal neurons. (**D**) Average traces of GCaMP6s epi-fluorescence (blue), hemodynamic signal (green reflectance), and hemodynamic-subtracted signal (red) surrounding spikes of single neurons, and random trigger average traces (black). The gray bands correspond to the standard deviation of random trigger average traces.

the limb sensorimotor areas. These results highlight the power of this multimodal approach, in that it can link high-resolution single neuron activity with network dynamics of the entire cortex and concurrent animal behavior, which greatly improves our ability to dissect the functional role of individual neurons of any brain region.

To quantify the distribution of distinctively patterned STMs of neurons across brain regions, we applied a graph-based clustering algorithm, Phenograph, on z-scored STMs (*Levine et al., 2015*; *Figure 5*). In total, we included 1146 STMs of 110 neurons recorded during multiple sessions. This resulted in 10 clusters with each cluster representing a distinct STM pattern evidenced by their within-cluster averaged map (*Figure 5A*). Un-matched STMs that were below the similarity threshold (Pearson's correlation <0.3) compared to any of the 10 cluster average maps were excluded (94 out of 1146 STMs). For all clusters, individual STMs within the clusters showed high similarity to the cluster average maps (*Figure 5B*). In order to validate the optimal number of clusters

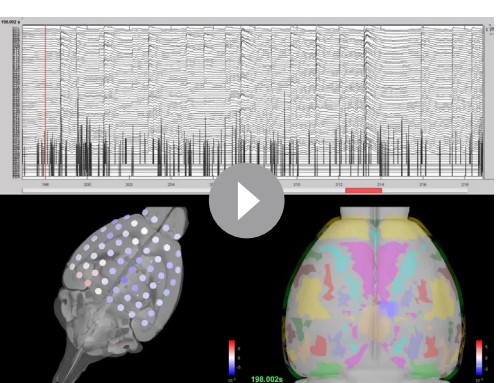

**Video 1.** Video of the 4D spatiotemporal whole-brain dynamics of mesoscale cortical calcium imaging and subcortical neuronal firing. Brain activity data was registered in a 3D mouse brain model. Calcium dynamic intensity and neuronal firing rate of subcortical neurons (Top) were color-coded in the ROIs in the 3D mouse brain model (bottom left). The calcium dynamic was overlaid on a cortical atlas (bottom right).
https://elifesciences.org/articles/87691/figures#video1

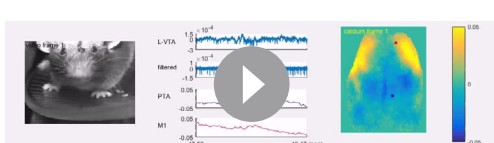

**Video 2.** Simultaneous recording of behavior video (left), L-VTA neuronal firing (middle), and wide-field calcium imaging (right) in a behaving mouse.
https://elifesciences.org/articles/87691/figures#video2

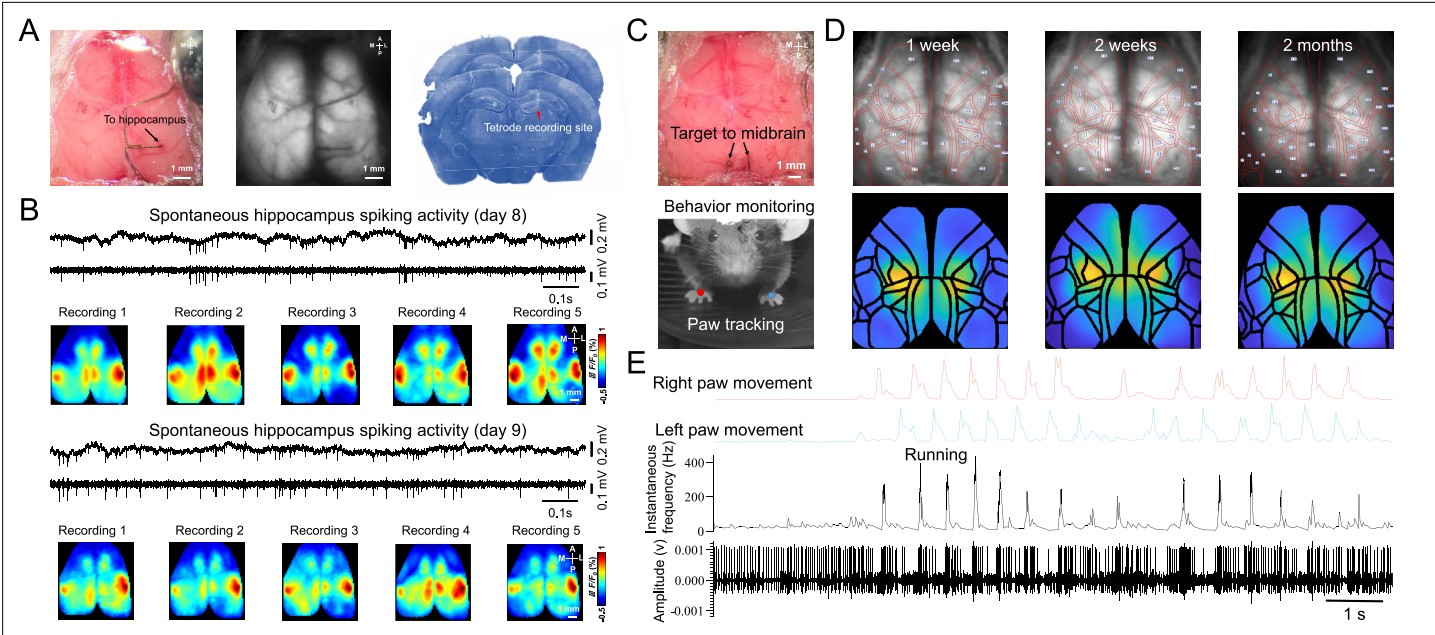

**Figure 4.** Chronic spike-triggered average maps of hippocampus and midbrain neurons. (**A**) Unfiltered brain RGB image showing the position of the tetrode (left), GCaMP6 fluorescence image (middle), and histology showing the recording site of the tetrode (right). (**B**) Example raw and high-pass filtered (>300 Hz) electrophysiological traces of hippocampal recording 7 days after tetrode implantation and STMs of the hippocampal neuron across recording sessions(top). Another day of hippocampal recording and STMs of the hippocampal neuron across recording sessions(bottom). (**C**) Unfiltered brain RGB image showing the position of the tetrode (top) and behavior video recording (bottom). (**D**) Top: Ca²⁺ images recorded 1 week (left), 2 weeks (middle), and 2 months (right) after tetrode implanted in the midbrain overlaid with automatically registered brain atlas using Mesonet. Bottom: STMs of the midbrain neuron recorded at the same time points as Top. (**E**) Synchronized traces of paw movements, instantaneous firing rate, and electrophysiological signals of the midbrain neuron are shown in (**C**).

for partitioning our dataset, we evaluated the clustering effectiveness of a varied number of clusters when applying K-means clustering algorithm using Silhouette score (*Rousseeuw, 1987*) and confirmed that 10 clusters are most appropriate (*Figure 5C*). Degrees of separation between clusters were visually inspected by projecting dimensionally reduced STMs onto 3D space using t-distributed stochastic neighbor embedding (t-SNE), a commonly used statistical method for plotting high-dimensional data points onto 3D space (*Figure 5D*).

*Figure 5E* shows the proportions of STMs of neurons in different brain regions that were assigned to each cluster. We found that STMs in each region exhibited a stereotypical distribution pattern. For example, STMs of cortical neurons were primarily assigned to clusters #1–3 and #5–7, corresponding to motor, barrel, and primary somatosensory cortex activation patterns. These activation areas also matched the locations of the recorded cortical neurons (e.g. motor cortical neurons showed motor area activated STMs), suggesting that the functional connectivity patterns of cortical neurons were localized to cortical areas surrounding the recorded neuron, which is consistent with previous reports (*Xiao et al., 2017*; *Clancy et al., 2019*). Hippocampal neurons primarily displayed STMs with retrosplenial cortex activation (cluster#10, 59.6% of all hippocampal neuron STMs), reaffirming the dense functional connectivity between these two regions (*Karimi Abadchi et al., 2020*). STMs of striatal neurons were dominated by two opposing patterns, where 47.7% of them showed a global activation except for the body (limb and trunk) somatosensory regions (cluster#4) and 28.8% of them showed only activation in the body sensory regions (cluster#9). Thalamic neurons exhibited diverse STM patterns with the primary activated regions in lateral anterior cortical regions (cluster#5, 5.6%, cluster#6, 23.3%, cluster#7 26.3% of all thalamic STMs) and medial cortical regions (cluster#8, 6.5%, cluster#9, 11.6%, cluster#10, 9.9% of all thalamic STMs). This suggested that thalamic neurons had diverse functional connectivity patterns in relation to the dorsal cortex, consistent with previous studies (*Xiao et al., 2017*). STMs of midbrain neurons were most assigned to cluster#8 with medial frontal and somatosensory cortex activation (cluster#8, 44.6% of all midbrain STMs), but also included functional connectivity patterns with various other cortical regions (cluster#6, 10.5%, cluster#9, 12.6%,

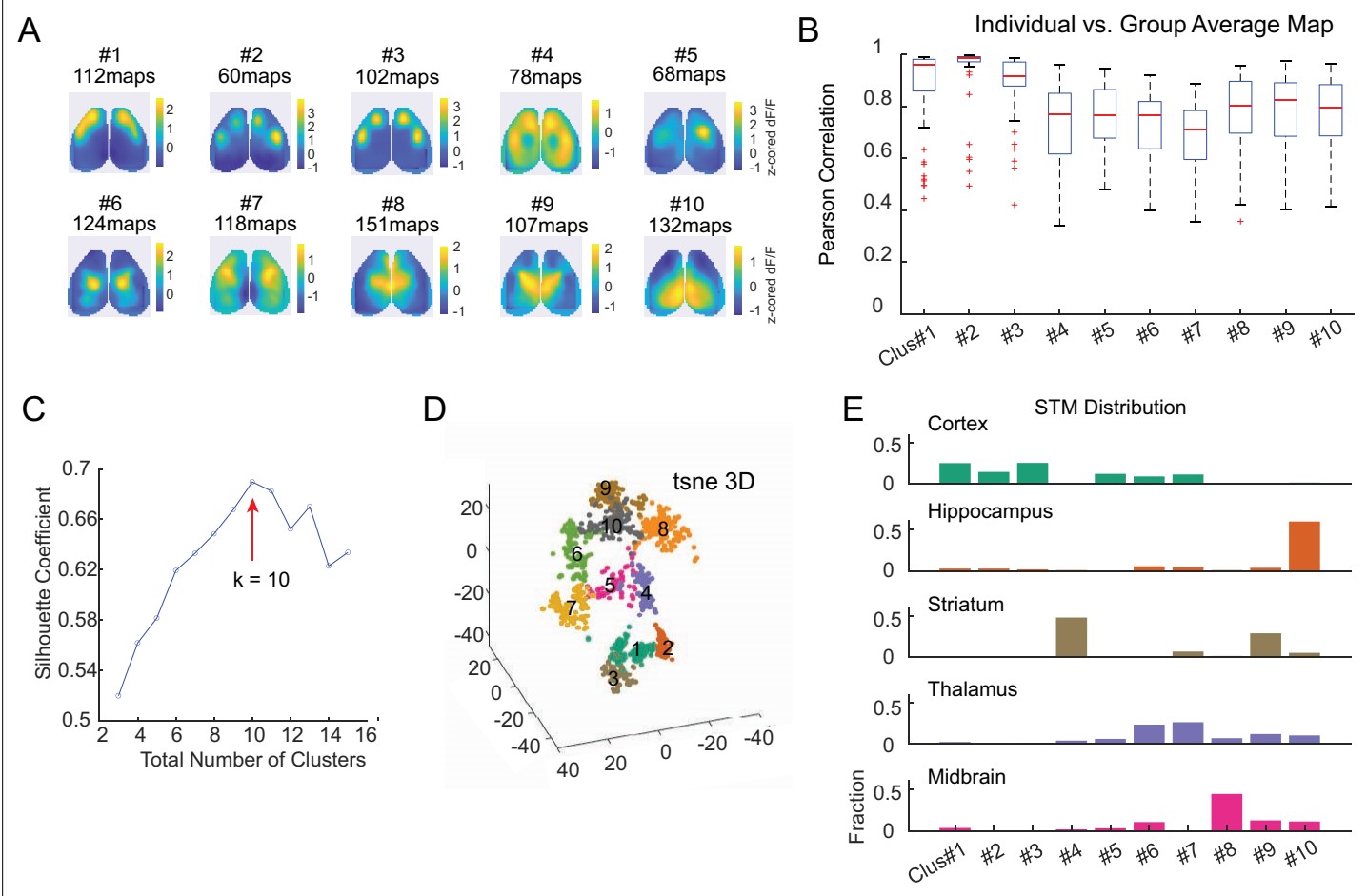

**Figure 5.** Clustering STMs of cortical/subcortical neurons. (**A**) Average z-scored STM of each cluster. Each cluster represents a unique cortical functional network that individual neurons were associated with. STMs of the same neurons recorded during different sessions were included (1146 total STMs from 110 neurons of 14 mice). (**B**) Box plot showing the correlation between individual STM and average STM of its assigned cluster. (**C**) Silhouette coefficient of varying numbers of clusters to validate the optimal number of clusters. (**D**) 3D-t-SNE plot showing dimensionality reduced representation of clustered individual STMs. (**E**) The proportion of neurons in each cortical and subcortical region assigned to each STM cluster.

cluster#10, 11.2% of all midbrain STMs). Interestingly, we found that neurons of different subcortical origins sometimes have overlapping STMs (e.g. cluster#9, and #10 exhibited by neurons from all subcortical regions), which suggests that they may be involved in the same subcortico-cortical functional pathway (*Figure 5E*). Taken together, these findings highlighted the capability of Mesotrode to capture the diverse cortical functional connectivity patterns of neurons across brain regions using chronic electrodes.

## Linkage of spiking peripheral nerve to specific mesoscopic cortical maps

One key advantage of using our minimally invasive micro-electrode setup is the flexibility of the location where they can be implanted and the relatively small footprint at the insertion site. This allows us to explore the functional connectivity of neurons that are otherwise difficult to study. For example, it's difficult to simultaneously record the neural activity of the facial motor nucleus and cortex in order to investigate the cortico-brainstem motor pathways that directly innervate facial muscles that control whisker movement and facial expressions in mice (*Petersen, 2014*; *Sreenivasan et al., 2016*; *Mercer Lindsay et al., 2019*). Here, we implanted micro-wires on the facial motor nerve to record the spiking activity of the axonal projections from the facial motor neurons (*Figure 6A and B*). We obtained the STMs of facial motor nerves by simultaneous wide-field imaging of GCaMP6 mice (n=5 mice). Surprisingly, we found a distinct cortical pattern, activation in RSP, M2, and PTA areas (*Table 1*),

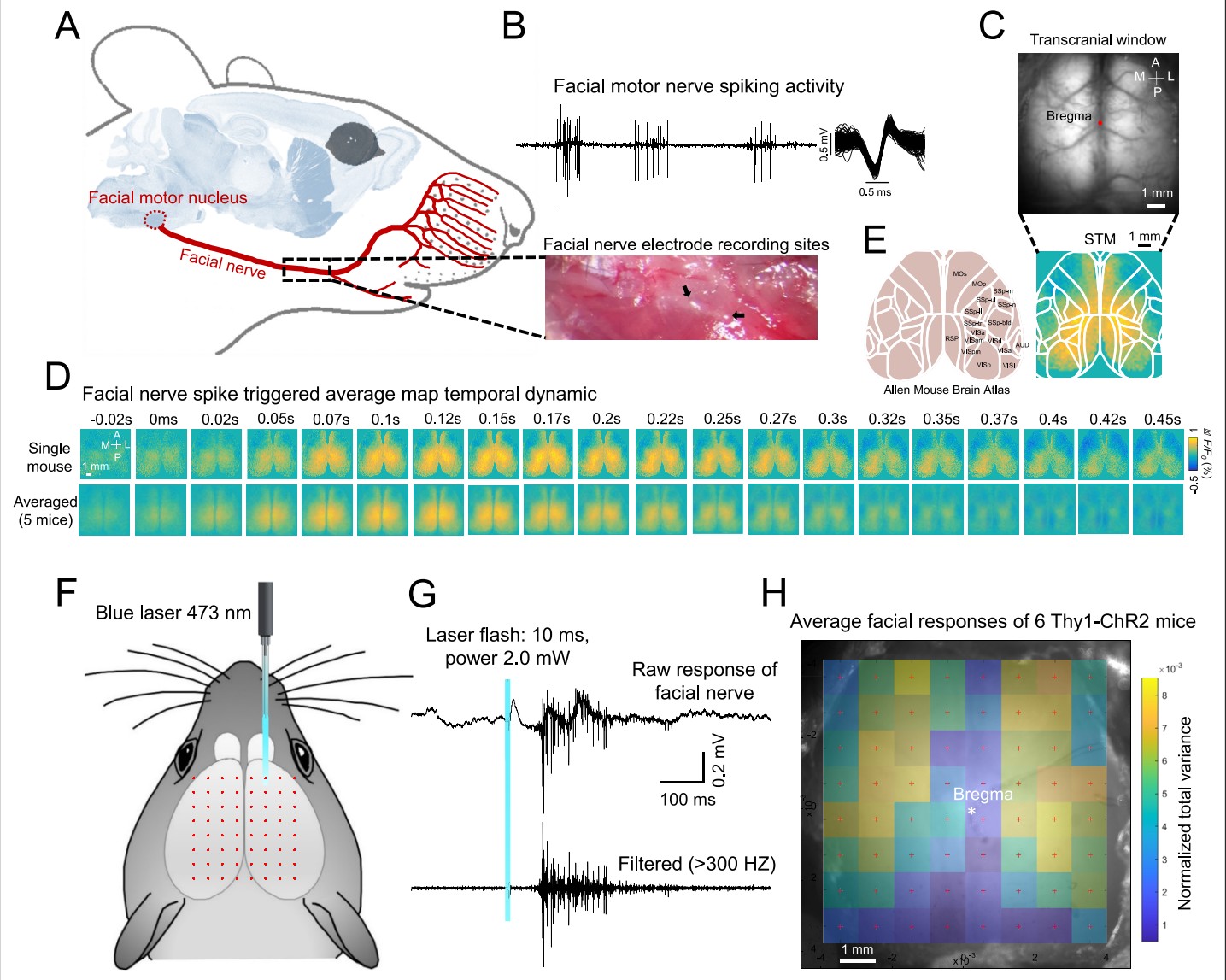

**Figure 6.** Spike-triggered and optogenetics mapping facial nerve spiking associated cortical maps. (**A**) Mouse anatomy illustrates facial motor nucleus axons (facial nerve) that innervate facial muscles. Micro-wire was implanted in the facial nerve. (**B**) Example spiking activity recorded from facial motor nerve using the micro-wire. (**C**) Example calcium imaging during facial nerve recording. (**D**) Facial nerve spike-triggered average map temporal dynamics surrounding facial nerve spikes (t=0ms) of a single mouse (top row) or the average of 5 mice (bottom row). (**E**) Reference atlas (white outlines; 2004 Allen Institute for Brain Science. Allen Mouse Brain Atlas. Available from: http://mouse.brain-map.org/) (**F**) Illustration of automated light-based mapping of the dorsal cortex (8x8 grid) by photoactivation of Thy1-ChR2 mice. (**G**) Example single trial facial nerve response after light stimulation. (**H**) Average change of facial nerve responses of 6 mice represented on a 2D color-coded map corresponding to each stimulation coordinate overlayed on an example cortical image of Thy1-ChR2 mouse. The amplitude of the average change for each stimulation coordinate corresponds to the total variance of all the trials (>5 trials, within 200ms after stimulus) where there is no facial nerve spiking before the stimulus.

The online version of this article includes the following figure supplement(s) for figure 6:

**Figure supplement 1.** Facial nerve spike-triggered average maps of 5 mice.

**Figure supplement 2.** Optogenetic mapping of facial nerve responses (blue laser, 10ms, 2.0 mW) in a Thy1-ChR2 mouse.

**Figure supplement 3.** Optogenetic mapping of facial nerve responses (blue laser, 10ms, 2.0 mW) in a Thy1-ChR2 mouse.

**Figure supplement 4.** Optogenetic mapping of facial nerve responses (blue laser, 10ms, 2.0 mW) in a Thy1-ChR2 mouse.

**Figure supplement 5.** Optogenetic mapping of facial nerve responses (blue laser, 10ms, 2.0 mW) in a Thy1-ChR2 mouse.

**Figure supplement 6.** Optogenetic mapping of facial nerve responses (blue laser, 10ms, 2.0 mW) in a Thy1-ChR2 mouse.

**Figure supplement 7.** Optogenetic mapping of facial nerve responses (blue laser, 10ms, 2.0 mW) in a Thy1-ChR2 mouse.

**Table 1.** Abbreviation used to define different cortical/sub-cortical areas.

| Mop (M1) | Primary motor area |
| --- | --- |
| Mos (M2) | Secondary motor area |
| SSp-m | Primary somatosensory area, mouse |
| SSp-ul | Primary somatosensory area, upper limb |
| SSp-ll | Primary somatosensory area, lower limb |
| SSp-n | Primary somatosensory area, nose |
| SSp-bfd | Primary somatosensory area, barrel field |
| SSp-tr | Primary somatosensory area, trunk |
| VISp | Primary visual area |
| VISa | Anterior visual area |
| VISam | Anteromedial visual area |
| VISpm | Posteromedial visual area |
| VISrl | Rostrolateral visual area |
| VISal | Anterolateral visual area |
| VISl | Lateral visual area |
| RSP | Retrosplenial area |
| AUD | Auditory areas |
| PTA | Parietal association area |
| HPF | Hippocampal formation |
| VTA | Ventral tegmental area |

associated with facial motor nerve spiking activity (*Figure 6C, D and E*, *Figure 6—figure supplement 1*, *Video 3*) in awake mice showing spontaneous movements. Using automated light-based mapping technology (*Ayling et al., 2009*), we found a broad area of the dorsal cortex, including the PTA region could evoke facial motor nerve spiking activity when optogenetically stimulated (*Figure 6F, G and H*, *Figure 6—figure supplements 2–7*, *Video 4*) within Thy1-ChR2 mice (n=6 mice). To further investigate causal relationships, we directly inhibited the PTA by optogenetically stimulating inhibitory neurons of the PTA region of VGAT-ChR2 mice (n=3 mice). Echoing our previous finding, inhib-

iting the PTA not only stopped the PTA neurons from firing but also prevented the facial movements of these mice when the blue light stimulation was on (*Figure 7*, *Video 5*). To eliminate the possibility that this observation is due to indirect inhibition of other brain areas, we simultaneously recorded neuronal activities in M1 and the barrel cortex (BCS). We confirmed that the neuron firing in these brain areas was not decreased by PTA

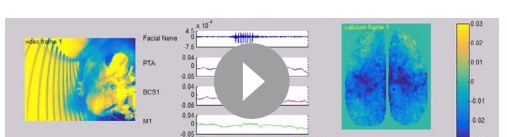

**Video 3.** Simultaneous recording of behavior video (left), facial nerve spiking activity (middle), and wide-field calcium imaging (right) during mouse whiskering.
https://elifesciences.org/articles/87691/figures#video3

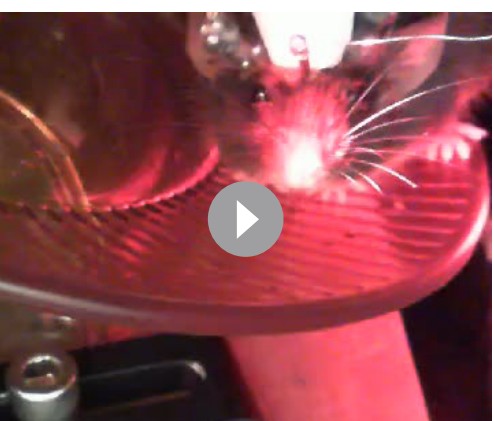

**Video 4.** Video of the facial response after optogenetic stimulation (blue laser, 10ms) of PTA in a Thy1-ChR2 mouse.
https://elifesciences.org/articles/87691/figures#video4

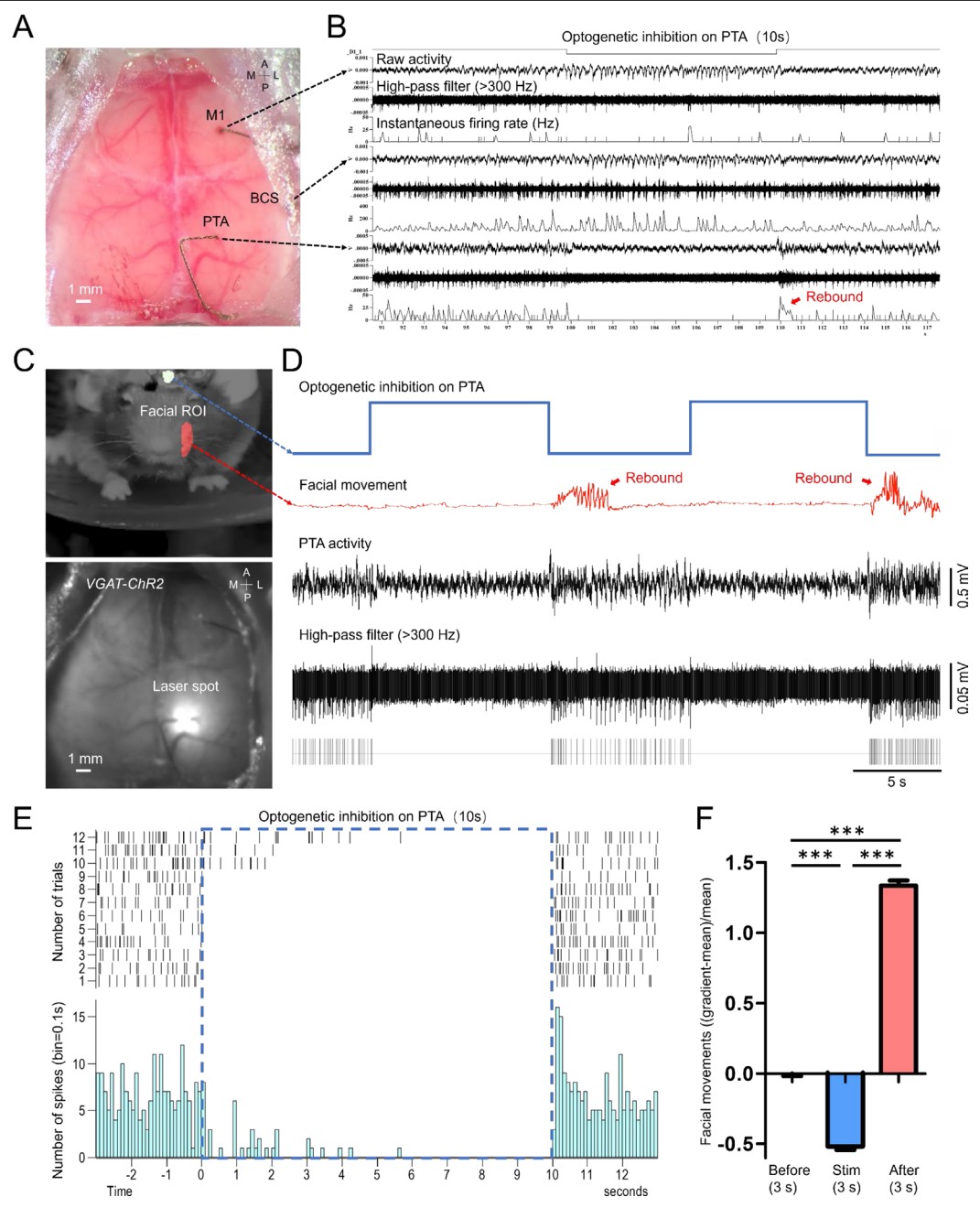

**Figure 7.** Optogenetic inhibition of the PTA area significantly reduced the facial movement of the mouse.
(**A**) Unfiltered brain RGB image showing the position of the tetrodes. (**B**) Simultaneous cellular electrophysiological recording in the cortical regions of M1, BCS, and PTA with optogenetic inhibition on the PTA region. (**C**) Behavioral image showing masks used for facial movement detection (top). Brain image of a VGAT-ChR2 mouse (ChR2 expressed in cortical inhibitory neurons) with laser targeting the PTA area (bottom). (**D**) Synchronized traces of optogenetic activation of PTA inhibitory neurons, facial movement, and LFP of the PTA region. Inhibiting the PTA region strongly inhibited facial nerve activity. Cessation of PTA inhibition reliably induced rebound PTA neuron firing and facial movements. (**E**) The neuronal firing of 12 trials of optogenetic inhibition in PTA. (**F**) Facial movements ((gradient-mean)/mean) 3 s before, 3 s during, and 3 s immediately after optogenetic inhibition in PTA for 29 trials (Dunn's Multiple Comparison Test, $p < 0.001$, n=3 mice).

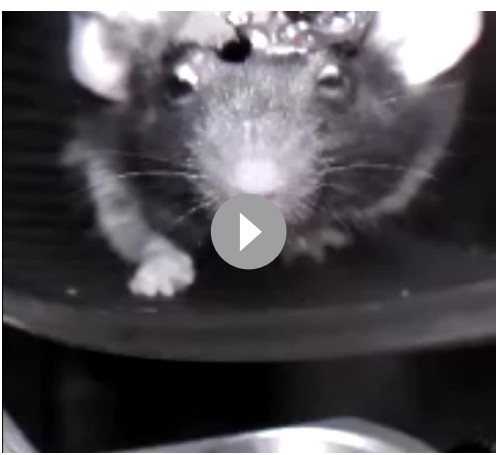

**Video 5.** Video of the facial response after optogenetic inhibition (blue laser, 10 s) of PTA in a VGAT-ChR2 mouse.

https://elifesciences.org/articles/87691/figures#video5

inhibition (*Figure 7A and B*). More interestingly, we saw a 'rebound effect', a sudden increase in PTA neuron firing and facial movement, right after the stimulation is turned off (*Figure 7C and D*). This 'off response' supports the causal relationships between PTA and facial movement. These findings are consistent with the previous report that the PTA is involved in controlling active movements (*Chapman et al., 2002*; *Cohen and Andersen, 2002*; *Rathelot et al., 2017*; *Auffret et al., 2018*; *Lyamzin and Benucci, 2019*; *Lee et al., 2022*). It also underscores that our chronic tetrode implants in combination with optogenetics and optical imaging can be a powerful tool in mapping and identifying novel functional pathways.

## Discussion

How activities of individual neurons contribute to brain-wide population dynamics remains poorly understood (*Brecht et al., 2004*; *Rancz et al., 2007*; *Houweling and Brecht, 2008*; *Packer et al., 2015*). It is additionally challenging to study long-range connectivities (e.g. those between subcortical and cortical structures) given the extraordinary complexity of neuronal wiring between brain regions. Non-invasive methods such as fMRI have generated important connectivity models, such as the default mode network (DMN) model, which advanced our understanding of how distinct brain regions communicate with each other during different behavioral contexts (*White et al., 2011*; *Gutierrez-Barragan et al., 2022*). However, these methods typically lack the spatial and temporal resolution to understand the connectivity patterns of individual neurons. On the other hand, for more invasive techniques such as viral tracing, although they often provide detailed characterizations of the anatomical patterns between brain areas, the nature of the preparation for these techniques precludes gaining insights into the dynamic nature of the connectivity rules. In this study, we developed the Mesotrode that combined chronic tetrode recording with mesoscale Ca$^{2+}$ imaging to characterize mesoscale connectivity patterns of individual neurons. We showed that the functional connectivity patterns of individual neurons are relatively stable in the same behavioral state, and neurons from different brain regions display distinct distributions of these patterns. We further showcased the utility of our procedure by recording from the facial motor nerve and found a distinct cortical activation associated with facial nerve activity. Additionally, Mesotrode provides a wide-field access window for optogenetic stimulations. We showed a causal link between PTA activity and facial movement in mice. These findings established that our multimodal recordings combining chronic electrophysiology, mesoscale imaging, optogenetics, and behavioral characterization can be widely applied to further our understanding of neuronal connectivity and behaviors.

### Compared to other chronic recording setups

Our work employs compact tetrode electrodes that permit simultaneous resolution of single units and mesoscale GCAMP activity. It is conceivable that improvements in spike sorting fidelity could be made by switching to more densely spaced silicon probes. While this is an obvious advantage, these probes do not have a compact footprint and would interfere with regional imaging. Our approach adds to the plethora of techniques previously described to characterize long-range functional connectivity by combining single-neuron activity recording and mesoscale cortical imaging. These techniques typically differ in how and where the individual neurons' activities are recorded. For instance, Barson et. al used two-photon imaging to record individual cortical neurons' activities, which allows differentiation of different neuronal types through genetic manipulation (*Barson et al., 2018*), but it has lower temporal resolution and is restricted to recording cortical neurons. Xiao et.al, Clancy et.al, Peters et. al, and Liu et.al all used either silicon probes or neuropixel probes to record large populations of

cortical or subcortical neurons (*Xiao et al., 2017*; *Clancy et al., 2019*; *Liu et al., 2021b*; *Peters et al., 2021*), but it remains difficult to use these setups to track activities of the same neurons over days. In addition to these approaches, we can use our chronic tetrode implantation to track neuronal activities at multiple brain regions over long periods of time to better describe the dynamic nature of global neuronal affiliations; however, the number of neurons that can be recorded per tetrode is limited which does reduce our data acquisition throughput.

Compared to most of the previously reported chronic tetrodes setups with the inclusion of a micro-drive, our approach has the unique advantages that setting up the apparatus and implanting the tetrodes is a lot more straightforward, mini-connector enables group housing and it is compatible with overhead cameras which permits simultaneous optical interrogation of neuronal circuits (*Battaglia et al., 2009*; *Nguyen et al., 2009*; *Voigts et al., 2013*; *Billard et al., 2018*; *Delcasso et al., 2018*; *Voigts and Harnett, 2020a*). However, it does come with the price that the tetrode cannot be moved after it is implanted, which means that the implant would fail if the health of the neuron deteriorates too much over time. Albeit, since our goal is to record the activity of the same neurons, we deemed it a worthy tradeoff for our purpose. Future studies could include experiments that would take advantage of bilateral symmetry by placing high-resolution silicon probes in one hemisphere and then reading out mesoscale maps in the other.

## Variability of spike-triggered average maps in different brain regions

Activities of cortical pyramidal neurons are known to be coupled to local network activity; as such, STMs of these neurons often display activity motifs localized to the cortical regions they reside, indicating that their functional connections are mostly involved in local networks (*Xiao et al., 2017*; *Clancy et al., 2019*). On the other hand, subcortical neurons (i.e. thalamic neurons) can display more diverse and distal connectivity patterns with cortical networks that are less confined by their locations (*Xiao et al., 2017*). Here, we extended these characterizations to more subcortical regions (i.e. striatum, hippocampus, and midbrain), and found that these neurons, similar to thalamic neurons, generate functionally diverse STMs. This can be attributed to the diverse cell types within these structures (*Phillips and Irvine, 1979*; *Cembrowski et al., 2016*). Furthermore, by classifying these STMs, we found that the connectivity patterns of neurons in different subcortical structures are varied. For instance, STMs of midbrain neurons can be primarily categorized into 4 classes, with each class having distinct activity motifs, which is in contrast to the striatum neurons, the STMs of which are grouped into only two main classes. More interestingly, STMs of different subcortical regions display low levels of overlap, suggesting that neurons of different subcortical origins form their own distinct connectivity pattern to the cortex.

## The spike-triggered average map associated with specific brain function

The single neuron defined functional maps (STMs) can reveal functional cortical architecture related to the activity of individual cortical and subcortical-cortical neurons (*Xiao et al., 2017*). The distinct STM patterns indicate that single neuron embedded large-scale subcortico-cortical networks may be associated with specific brain functions (*Pessoa, 2014*; *Liu et al., 2021a*; *Amunts et al., 2022*). To explore this possibility, for the first time, we simultaneously record the output spiking activity of the facial motor nucleus and wide-field calcium dynamics of the cortex, which enable us to investigate the cortico-brainstem motor pathways that directly control facial movements in mice. Interestingly, we found that facial nerve spike-triggered average map functionally associated with active facial movements, and the causal relationship can be tested by optogenetic stimulation or inhibition of the PTA area. Previous studies using anterograde virus tracing have revealed that the motor cortex sends few direct projections to the facial nucleus while the most prominent projections from the motor, somatosensory, and association cortex are polysynaptic, presumably through nuclei in the brainstem (*Grinevich et al., 2005*; *Matyas et al., 2010*; *Rathelot et al., 2017*). Studies in humans and monkeys have identified parietal-eye-field and reach-specific areas that involve voluntary control of eye or hand movement (*Lynch, 1980*; *Andersen, 1989*; *Sakata et al., 1997*; *Chapman et al., 2002*; *Konen and Kastner, 2008*; *Archambault et al., 2009*; *Pouget, 2015*; *Rathelot et al., 2017*). Our findings suggest that the PTA region is also involved in facial movements in mice, and supplement the current understanding of the functional connectivity between cortex and facial nucleus motor neurons.

## Conclusion

Our results demonstrated that our minimally invasive, flexible, low profile, low cost, chronically implanted Mesotrode can be readily applied to interrogating the functional connectivity between single neurons and large-scale cortical networks, and the causal relationship can be tested by opto-genetic stimulation.

# Materials and methods

## Mice

Animal protocols (A13-0336-A22-0054 and A14-0266-A22-0258) were approved by the University of British Columbia Animal Care Committee and conformed to the Canadian Council on Animal Care and Use guidelines and animals were housed in a vivarium on a 12 h daylight cycle (7 AM lights on). Most experiments were performed toward the end of the mouse light cycle. Transgenic GCaMP6s mice (males, 2–4 months of age, weighing 20–30 g; n=19) were produced by crossing Emx1-cre (B6.129S2-*Emx1*$^{tm1(cre)Krj}$/J, Jax #005628), CaMK2-tTA (B6.Cg-Tg(Camk2a-tTA)1Mmay/DboJ, Jax #007004) and TITL-GCaMP6s (Ai94;B6.*Cg-Igs7*$^{tm94.1(tetO-GCaMP6s)Hze}$/J, Jax #024104) strain. The presence of GCaMP expression was determined by genotyping each animal before each surgical procedure with PCR amplification. These crossings are expected to produce a stable expression of the calcium indicator (*Chen et al., 2013b*) specifically within all excitatory neurons across all layers of the cortex (*Vanni and Murphy, 2014*). Channelrhodopsin-2 transgenic mice (*Arenkiel et al., 2007*) (n=6) were obtained from the Jackson Labs (line 18, stock 007612, strain B6.Cg-Tg(Thy1-COP4/EYFP)18Gfng/J). Neuronal activity was driven by optically activating inhibitory neurons targeted in VGAT-ChR2-YFP expressing mice (*Zhao et al., 2011*; n=3). No method of randomization was used since all mice belonged to the same sample group. Samples sizes were chosen based on previous studies using similar approaches (*Mohajerani et al., 2013*; *Vanni and Murphy, 2014*). Given the use of automated acquisition and analysis procedures, we did not employ blinding.

## Tetrode fabrication

Polyimide insulated, tungsten wire (40 cm long, diameter 25 µm, Stablohm 650, California Fine Wire, USA) was folded into four wires and clamped together with the modified alligator clip. The loop of wires was hung over the horizontal bar. The alligator clip was placed into the motorized stage. Eighty clockwise twists were applied and followed by forty counter clock-wise twists to the wire bundle over the course of approximately three minutes. After tetrode twisting was completed, the wires were fused together by heating from three different angles with a heat gun (420 °C or 790 °F), using medi-um-low flow. For each angle, begin 1–2 cm below where the wire bundle splits, the heat gun was held 2 cm away from the wire and moved periodically over the length of the wire for about 5 s. The tetrode was removed from the twisting apparatus by gently lifting the alligator clip to relieve tension on the tetrode, and cutting the tetrode near the alligator clip. At the other end, the loop was cut off such that there were four non-bonded strands of wire of equal length. The tetrode was then soldered to a miniature connector using a fine-tip soldering iron (Thermaltronics M8MF375 Micro Fine 0.25 mm).

## Surgical procedures

Mice were anesthetized with isoflurane (4% for induction and 1.5% for maintenance in air). The eyes were covered with eye lubricant (Lacrilube; https://www.well.ca/) and body temperature was main-tained at 37 °C with a heating pad with feedback regulation. Mice were then placed in a stereotaxic frame and received an injection of Lidocaine (0.1 ml, 0.2%) under the scalp. Mice also received a 0.5 ml subcutaneous injection of a saline solution containing buprenorphine (2 mg/ml), atropine (3 µg/ml), and glucose (20 mM). Respiration rate and response to toe pinch were checked every 10–15 min to maintain the surgical anesthetic plane. A skin flap extending over both hemispheres approximately 8 mm in diameter (3 mm anterior to bregma to the posterior end of the skull and down lateral to eye level) was cut and removed. Fascia or connective tissue is lightly scraped away from the skull and small (<1 mm diameter) holes are drilled through the skull, using a high-speed dental drill with a sterile bit, over the cortex. Tetrode was directed toward the center of the hole and placed in the target using a motorized micromanipulator (MP-225, Sutter Instrument Company). Miniature connectors (2x2 x 2

mm) are cemented to the skull (with dental adhesive) around the imaging window (the total weight is <1 gm). Ground and reference electrodes are fixed into place on the surface of the posterior skull.

After tetrode implantation, the chronic transcranial window was implanted as previously described (*Silasi et al., 2016*). Briefly, the skin between the ears and the eyes was properly cleaned with Betadine dissolved in water and ethanol. The skin covering the occipital, parietal, and frontal bones was cut away. The fascia and any connective tissue on the skull surface were gently wiped off. C&B-Metabond transparent dental cement (Parkell, Edgewood NY, USA; Product: C&B Metabond) was prepared in a ceramic bowl and used to glue a head-fixing titanium bar to the cerebellar plate or a 4/40 stainless steel setscrew slightly angled posteriorly (~120° relative to skull). With the bar or the setscrew in place, a thick layer of dental adhesive was applied to the skull. A coverglass (Marien-feld, Lauda-Konigshofen, Germany; Cat n: 0111520) previously cut to the size of the final cranial window (~8 mm diameter), was placed on top of the dental cement before it solidified, preventing the formation of bubbles. The cement remains transparent after it solidifies and the surface vasculature should be readily seen through the final result.

For facial nerve recordings, peripheral nerve activity was measured by fine wire recording directly from the nerves subserving the whisker. During surgery, mice will be anesthetized and positioned on a warming pad connected to a rectal probe and temperature maintained at 37 °C. A skin incision was made, exposing a small part of the buccal branch of the left facial nerve. Magnification of the surgical field with a dissecting microscope allowed a careful dissection of a nerve branch with minimum disruption of the tissues and blood supply surrounding the nerve. The appropriate site of exposure was determined by using two projection lines: a vertical line running downward, posterior from the outer corner of the eye, and a horizontal line running in caudal direction, starting at the whisker E-row. Then two insulated fine wires (about 25 µm tips) were hooked and placed around the nerve separated about 2 mm from one another. The insulation at the ends of the wires was removed and a knot was made on each wire to prevent it from slipping. The opposite ends of each wire were soldered to a mini connector attached by dental cement to the skull. Finally, 6–0 silk sutures were used to close the skin incisions.

## Recovery and post-operative monitoring

At the end of the surgical procedures, mice received a subcutaneous injection of saline (0.5 ml) with 20 mM of glucose and were allowed to recover in their home cages with an overhead heat lamp. The activity level of mice that underwent the procedure was monitored hourly for the first 4 hr and every 4–8 hr thereafter. Mice are allowed to recover for 7 days from the window implantation before performing electrophysiology recording.

## Electrophysiology data acquisition

For Electrophysiology recordings, a custom adapter was connected to the miniature connector. The tetrode signals were amplified using a 16-channel data acquisition system (20 kHz, USB-ME16-FAI-System, Multi-Channel Systems) and recorded for at least 30 min every day.

Raw extracellular traces were imported into Spike2 (Cambridge Electronic Design, Cambridge, UK) or SpikeSorter software (*Swindale and Spacek, 2014*; *Swindale et al., 2021*) for spike sorting and analysis. Briefly, data were high pass-filtered at 1 kHz, and excitatory spikes were detected using a threshold of 4.5 times the median of the standard deviation over 0.675. Sorting was carried out by an automated method previously described (*Swindale and Spacek, 2014*) and followed by manual visual inspection of units.

## Wide-field calcium imaging

All mice were habituated for 1 week prior to data collection. Awake mice were head-fixed and placed on a wheel in a dark imaging chamber for data collection. A behavioral camera (Raspberry Pi camera) and an infrared light were placed inside the imaging chamber to monitor active behaviors, such as running or whisking. A Pantera 1M60 CCD camera (Dalsa) was equipped with two front-to-front lenses (50 mm, f ¼ 1.435 mm, f ¼ 2; Nikon Nikkor) and a bandpass emission filter (525/36 nm, Chroma). The 12-bit images were captured at a frame rate of 120 Hz (exposure time of 7ms) with 8×8 on-chip spatial binning using EPIX XCAP v3.8 imaging software. The cortex was sequentially illuminated with alternating blue and green LEDs (Thorlabs). Blue light (473 nm) with a bandpass filter (467–499 nm)

was used to excite calcium indicators and green light (525 nm) with a bandpass filter (525/50 nm) was used to observe changes in cerebral blood volume. The blue and green LEDs were sequentially activated and synchronized to the start of each frame's exposure period with transistor-transistor logic such that each frame collected only fluorescence or reflectance signals at 60 Hz each. This LED strobe frequency of 60 Hz exceeded the critical flicker fusion frequency for mice, which marks the highest temporal frequency that an observer can resolve flicker before it becomes indistinguishable from constant light and was likely imperceptible to the mice. Reflectance signals were subtracted from fluorescence signals to mitigate the contribution of hemodynamics to fluorescence signals.

## Optogenetic stimulation protocols

Inhibitory neurons in the cortex were optogenetically manipulated separately in VGAT-ChR2 mice. A 473 nm laser (Crystal Laser BCL-473–050, Reno, NV, USA) was connected to a fiber optic patch cable (0.22 NA, 200 µm gauge; Thorlabs FG200UCC, USA). During the experiment, 1 mW (measured from the end of the cable) of optical stimulation was delivered for an overall time of 10 s. For automated Thy1-ChR2-based facial motor mapping, we chose a relatively collimated 473 nm laser targeted through a simple microscope. The laser was moved in random order to each of the predefined stimulation locations (8x8 grid) and delivered a flash of laser light (10ms, 2.0 mW) to each point while collecting facial motor nerve response.

## Spike-triggered average maps (STM) Clustering

We employed an unsupervised clustering algorithm, PhenoGraph (*Levine et al., 2015*), to categorize STMs. The graph was built in 2 steps: (1) it finds k nearest neighbors for N input vectors (4096 dimensions for 64x64 STMs) using Euclidean distance, resulting in N sets of k nearest neighbors, (2) a weighted graph is built where the edge weight between pairs of nodes depends on the number of neighbors they share. We then perform Louvain community detection (*Blondel et al., 2008*) on this graph to partition the graph that maximizes modularity. This algorithm only requires one input parameter which is k, the number of nearest neighbors to be found for each input vector, and the resulting number of clusters (10) is relatively insensitive to the k chosen over the range of typically chosen values. We also performed k-means clustering using the same input and compared clustering effectiveness using the Silhouette score and found that 10 clusters were optimal. We then performed a cluster identity refinement step, where an individual STM whose similarity (Pearson correlation value) with the cluster average STM of the assigned cluster was lower than 0.3 was manually reassigned if its similarity with group average STMs of any other clusters was higher than 0.3; otherwise, that STM was excluded. A total of 94 out of 1146 STMs (8.2%) were excluded.

## Histology

At the end of each experiment, animals were killed with an intraperitoneal injection of pentobarbital (24 mg). Mice were transcardially perfused with PBS followed by chilled 4% PFA in PBS. Coronal or sagittal brain sections (50 µm thickness) were cut on a vibratome (Leica VT1000S). Images of brain tissues with counter-stained DAPI were acquired using confocal microscopy (Zeiss LSM510) to reveal the electrode track and help identify the approximate cortical/subcortical location of recorded cells.

## Statistical analysis

Statistical analyses were performed using GraphPad Prism, Version 9. Statistical tests and results are reported in the figure legends. More than two groups were compared using one-way ANOVA followed by Dunnett's test comparing experimental flies to each control. All graphs represent mean ± SEM. Sample sizes are listed in the figure legends. No explicit power analyses were used to determine sample sizes prior to experimentation. Minimum sample sizes were decided prior to experimentation based on previous experience or pilot experiments revealing how many samples are typically sufficient to detect reasonable effect sizes.

## Acknowledgements

This work was supported by a Canadian Institutes of Health Research (CIHR) Foundation Grant FDN-143209 and project grant to THM and the UBC Institute of Mental Health Marshall Scholars and Fellowship program. THM was also supported by the Brain Canada Neurophotonics Platform,

a Heart and Stroke Foundation of Canada grant in aid, National Science and Engineering Council of Canada (NSERC; GPIN-2022–03723), and a Leducq Foundation grant. This work was supported by resources made available through the Dynamic Brain Circuits cluster and the NeuroImaging and NeuroComputation Centre at the UBC Djavad Mowafaghian Centre for Brain Health (RRID SCR_019086) and made use of the DataBinge forum. DX was supported by the Michael Smith Foundation for Health Research. We thank Pumin Wang, and Cindy Jiang for surgical assistance, Jamie D Boyd for technical assistance. We thank Hongkui Zeng and Allen Brain Institute for providing transgenic mice.

## Additional information

### Funding

| Funder | Grant reference number | Author |
| --- | --- | --- |
| Canadian Institutes of Health Research | PJT-180631 | Timothy H Murphy |
| Canadian Institutes of Health Research | FDN-143209 | Timothy H Murphy |
| Brain Canada Neurophotonics Platform | | Timothy H Murphy |
| Heart and Stroke Foundation of Canada | Grant in aid | Timothy H Murphy |
| Natural Sciences and Engineering Research Council of Canada | 2022-03723 | Timothy H Murphy |
| Leducq Foundation | | Timothy H Murphy |

The funders had no role in study design, data collection and interpretation, or the decision to submit the work for publication.

### Author contributions

Dongsheng Xiao, Conceptualization, Resources, Data curation, Software, Formal analysis, Investigation, Methodology, Writing – original draft, Writing – review and editing; Yuhao Yan, Data curation, Software, Formal analysis, Methodology, Writing – original draft, Writing – review and editing; Timothy H Murphy, Resources, Software, Supervision, Funding acquisition, Validation, Writing – original draft, Writing – review and editing

### Author ORCIDs

Dongsheng Xiao ⓘ http://orcid.org/0000-0002-1669-0021
Timothy H Murphy ⓘ https://orcid.org/0000-0002-0093-4490

### Ethics

Animal protocols (A13-0336-A22-0054 and A14-0266-A22-0258) were approved by the University of British Columbia Animal Care Committee and conformed to the Canadian Council on Animal Care and Use guidelines.

Reviewer #1 (Public Review): https://doi.org/10.7554/eLife.87691.2.sa1
Reviewer #2 (Public Review): https://doi.org/10.7554/eLife.87691.2.sa2
Author Response https://doi.org/10.7554/eLife.87691.2.sa3

## Additional files

### Supplementary files
• MDAR checklist

## Data availability

Code used for data analysis and data generated in this study have been deposited in Open Science Framework repositories https://osf.io/67p3s/ and https://osf.io/ke6fg/.

The following datasets were generated:

| Author(s) | Year | Dataset title | Dataset URL | Database and Identifier |
|---|---|---|---|---|
| Xiao D, Yan Y, Murphy TH | 2023 | Mesotrode | https://doi.org/10.17605/OSF.IO/67P3S | Open Science Framework, 10.17605/OSF.IO/67P3S |
| Xiao D, Yan Y, Murphy TH | 2023 | Mesotrode2 | https://doi.org/10.17605/OSF.IO/KE6FG | Open Science Framework, 10.17605/OSF.IO/KE6FG |

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
