## [Editor Report · eLife assessment]

This **valuable** study combines chronic widefield calcium imaging of dorsal cortex activity at the mesoscale level with electrical recording of single neurons in specific cortical and subcortical locations. This work provides **compelling** evidence for recording neuronal activity at multiple temporal and spatial scales by combination of optical and electrophysiological methods. This work will be of broad interest to system neuroscientists studying neural circuits.

---

## [Referee Report · Reviewer #1 (Public Review)]

The paper by Dongsheng Xiao, Yuhao Yan and Timothy H Murphy presents a timely approach to record neuronal activity at multiple temporal and spatial scales. Such approaches are at the forefront of system neuroscience and a few examples include, among others, fMRI alongside electrophysiology (Logothetis et al, 2021. Nature) or widefield calcium imaging (Lake et al, 2020. Nat Meth) , or functional ultrasound imaging and multi unit recording (Claron et al, 2023 Cell Reports), The method presented here combines "low resolution" (i.e. cortical regions) widefield calcium imaging across most of the dorsal portions of the murine cortex combined with electrical recording of single neurons in specific cortical and subcortical locations (as a matter of fact, this later components can be used everywhere in the murine brain).

The method presented here is straightforward to implement and very well documented. Examples of novel insights that this approach can generate are well presented and demonstrate the strength of the presented approach, some aspects of the analysis require clarification.

For example, the author reveal Spike-Triggered average cortical activation Maps (STMs) linked to the activity of single neurons (Figs 4 and 5) This allows to directly asses the functional connectivity between cortical and sub-cortical areas. It nevertheless unclear what is the stability of the established relationships. The nature of the "recordings" in Fig 4. is unclear. It looks like these are imaging sessions on the same day, the length of these recordings as well as the interval between them is not stated. It will be fundamental to build a metric to compare STMs variability across sessions/recordings/days; a root-mean-square from an average map across all recordings could provide a starting point.

Also with respect to the STMs analysis, the data-driven choice of 10 clusters might need a bit more explorations. While the silhouette clustering accuracy peaks at 10 (Fig 5A), this metrics comes without a confidence intervals making it difficult to know if a difference of less than 10% (i.e. 11 or 13 clusters) should be deemed different. Maybe a bootstrapping approach could be used here to build such confidence intervals. Another approach to reach the number of cluster to use could be based on "consensus" between different partitioning algorithms (e.g. Strehl, A. & Ghosh, J. itions. J. Mach. Learn. Res. 3, 583-617 [2001]). A much stronger argument should be provided to use the 0.3 correlation cutoff value which seems to be arbitrarily low. The main point here is that the authors should show that their conclusions hold within a range of parameter values (number of clusters and correlation threshold).

---

## [Referee Report · Reviewer #2 (Public Review)]

The article presents 'Mesotrode,' a technique that integrates chronic widefield calcium imaging and electrophysiology recordings using tetrodes in head-fixed mice. This approach allows recording the activity of a few single neurons in multiple cortical/subcortical structures, in which the tetrodes are implanted, in combination with widefield imaging of dorsal cortex activity on the mesoscale level, albeit without cellular resolution. The authors claim that Mesotrode can be used to sample different combinations of cortico-subcortical networks over prolonged periods of time, up to 60 days post-implantation. The results demonstrate that the activity of neurons recorded from distinct cortical and subcortical structures are coupled to diverse but segregated cortical functional maps, suggesting that neurons of different origins participate in distinct cortico-subcortical pathways. The study also extends the capability of Mesotrode by conducting electrophysiological recordings from the facial motor nerve. It demonstrates that facial nerve spiking is functionally associated with several cortical areas( PTA, RSP, and M2), and optogenetic inhibition of the PTA area significantly reduced the facial movement of the mice.

Studying the relationship between widefield cortical activity patterns and the activity of individual neurons in cortical and subcortical areas is very important, and Murphy's lab has been a pioneer in the field. However, the choice of low-yield recording methods (tetrode) instead of more high-yield recording techniques, such as silicon probes, makes the approach presented in this study somewhat less appealing. Also, the authors claim that a tetrode-based approach can allow chronic recordings of single neural activity over days - a topic that is very controversial. In terms of results, I was under the impression that most of the conclusions presented in the bulk of the paper ( Figures 1-5) are very similar to what previous work from Murphy's lab and other labs has shown using acute preparation. In this respect, the paper can benefit from a more in-depth analysis of the heterogeneity of single-neuron functional coupling. The last part of the facial nerve recording is interesting (Figure 6), but I think it can be integrated better into the rest of the paper.

---

## [Author Response]

We outline reviewer/editor queries, our responses are indicated below we thank the reviewers for their suggestions that we address below and with minor edits (that do not appreciably change the content such as figure lettering and methods information).

**Reviewer #1 (Public Review):**
The paper by Dongsheng Xiao, Yuhao Yan and Timothy H Murphy presents a timely approach to record neuronal activity at multiple temporal and spatial scales. Such approaches are at the forefront of system neuroscience and a few examples include, among others, fMRI alongside electrophysiology (Logothetis et al, 2021. Nature) or widefield calcium imaging (Lake et al, 2020. Nat Meth) , or functional ultrasound imaging and multi unit recording (Claron et al, 2023 Cell Reports), The method presented here combines "low resolution" (i.e. cortical regions) widefield calcium imaging across most of the dorsal portions of the murine cortex combined with electrical recording of single neurons in specific cortical and subcortical locations (as a matter of fact, this later components can be used everywhere in the murine brain).The method presented here is straightforward to implement and very well documented. Examples of novel insights that this approach can generate are well presented and demonstrate the strength of the presented approach, some aspects of the analysis require clarification.For example, the author reveal Spike-Triggered average cortical activation Maps (STMs) linked to the activity of single neurons (Figs 4 and 5) This allows to directly asses the functional connectivity between cortical and sub-cortical areas. It nevertheless unclear what is the stability of the established relationships. The nature of the "recordings" in Fig 4. is unclear. It looks like these are imaging sessions on the same day, the length of these recordings as well as the interval between them is not stated. It will be fundamental to build a metric to compare STMs variability across sessions/recordings/days; a root-mean-square from an average map across all recordings could provide a starting point.

Our goal was to present a well-documented protocol for implanting electrodes (tetrodes and peripheral nerve) that do not impede cortical mesoscale imaging and support chronic investigation of spike trains. We do provide examples of repeated spiking measurements across days from the same electrodes and animals. Unfortunately, due to the pandemic interrupting data collection and other factors, this dataset does not contain a thorough analysis of response longevity using these electrodes, but we do show examples in the figures. In Figure 1F, G, we showed that the single unit activity was relatively stable during one week, two weeks, and two months of recordings after implantation. In Figure 4B we showed spiking activity in the hippocampus was stable across day 8 and day 9. We also showed that the STM of the hippocampus neuron was consistently associated with the RSP, BCS, and M2 region for 10 recording sessions across days. In Figure 4D, We showed that the STMs of a midbrain neuron were relatively stable over 2 months. The spiking activity of the neuron on different days was consistently correlated with the lower limb, upper limb, and trunk sensorimotor areas on both hemispheres of the cortex.

Also with respect to the STMs analysis, the data-driven choice of 10 clusters might need a bit more explorations. While the silhouette clustering accuracy peaks at 10 (Fig 5A), this metrics comes without a confidence intervals making it difficult to know if a difference of less than 10% (i.e. 11 or 13 clusters) should be deemed different. Maybe a bootstrapping approach could be used here to build such confidence intervals. Another approach to reach the number of cluster to use could be based on "consensus" between different partitioning algorithms (e.g. Strehl, A. & Ghosh, J. itions. J. Mach. Learn. Res. 3, 583-617 [2001]). A much stronger argument should be provided to use the 0.3 correlation cutoff value which seems to be arbitrarily low. The main point here is that the authors should show that their conclusions hold within a range of parameter values (number of clusters and correlation threshold).

Thank you for the interesting suggestions regarding cluster numbers. We agree that the number (10 clusters) could be taken as an arbitrary value. However, we have done previous work examining cortical connectivity maps in Mohajerani et al. 2013 Nature Neurosci. and found that cortical mesoscale activity has a degree of freedom (number of unique elements) in the range of 10-15. This number is also supported by major structural networks found by the Allen Brain Connectivity Atlas and within functional imaging data. In other work using unsupervised methods Xiao et al. 2021 Nature Comm a similar number of clusters were identified so these numbers are without some basis.

**Reviewer #1 (Recommendations For The Authors):**
I enjoyed very much reading the manuscript!Minor comments (aesthetics and typos)Please clarify how the hemodynamic correction was performed. The text refers to "substracted". This usually involves the computation of a general of per-pixel weight. Is this correction constant along the longitudinal imaging session (i.e. over weeks)?

The hemodynamic correction was calculated based on the results of each daily session. Typically these corrections have minimal impact on overall values and are not expected to appreciably change over time.

In Figure 3, authors might reconsider scaling down the size of panel A and enlarging the data presented in D. Also, with respect to panel D, what does the gray band represent, confidence intervals, standard dev? Please clarify.

The gray bands correspond to the standard deviation of random trigger average traces.

Lines in 4E could be made thicker.In the caption of fig6, panel D is mentioned twice (should be E).

Thanks for catching this mistake we have changed the caption in the online version.

**Reviewer #2 (Public Review):**
The article presents 'Mesotrode,' a technique that integrates chronic widefield calcium imaging and electrophysiology recordings using tetrodes in head-fixed mice. This approach allows recording the activity of a few single neurons in multiple cortical/subcortical structures, in which the tetrodes are implanted, in combination with widefield imaging of dorsal cortex activity on the mesoscale level, albeit without cellular resolution. The authors claim that Mesotrode can be used to sample different combinations of cortico-subcortical networks over prolonged periods of time, up to 60 days post-implantation. The results demonstrate that the activity of neurons recorded from distinct cortical and subcortical structures are coupled to diverse but segregated cortical functional maps, suggesting that neurons of different origins participate in distinct cortico-subcortical pathways. The study also extends the capability of Mesotrode by conducting electrophysiological recordings from the facial motor nerve. It demonstrates that facial nerve spiking is functionally associated with several cortical areas( PTA, RSP, and M2), and optogenetic inhibition of the PTA area significantly reduced the facial movement of the mice.Studying the relationship between widefield cortical activity patterns and the activity of individual neurons in cortical and subcortical areas is very important, and Murphy's lab has been a pioneer in the field. However, the choice of low-yield recording methods (tetrode) instead of more high-yield recording techniques, such as silicon probes, makes the approach presented in this study somewhat less appealing. Also, the authors claim that a tetrode-based approach can allow chronic recordings of single neural activity over days - a topic that is very controversial. In terms of results, I was under the impression that most of the conclusions presented in the bulk of the paper ( Figures 1-5) are very similar to what previous work from Murphy's lab and other labs has shown using acute preparation. In this respect, the paper can benefit from a more in-depth analysis of the heterogeneity of single-neuron functional coupling. The last part of the facial nerve recording is interesting (Figure 6), but I think it can be integrated better into the rest of the paper.
**Reviewer #2 (Recommendations For The Authors):**
Major Comments:1. The methodology described in the paper is based on chronic tetrode recordings combined with widefield calcium imaging. The authors emphasize the advantages of using tetrodes in that they are (1) easy to implant (2) have a small footprint, and (3) allow to record the same neurons over days.I agree regarding the first advantage, however, the ability to reliably record the activity of the same neurons over days using electrophysiological recordings is controversial. The authors claim that:'We found that the single unit activity was relatively stable, during one week, two weeks, and two months of recordings after implantation (Figure 1F, G)',The only 'proof' the authors show for recording stability are waveforms of one neuron on one channel (out of presumably four channels), which seem to differ in amplitude over days. Two-dimensional plots of the neuron waveform for all channel combinations could be a more convincing way to make this claim. But, as I already mentioned - the ability to record from the same neurons chronically with electrophysiological methods is rather controversial, especially with tetrodes that don't allow for laminar profiling of neuronal response to account for a potential drift over time.

We now make it more clear that examples of mesotrode stability are indicated in the figures. Furthermore, we acknowledge caveats that spike sorting experiments required to more conclusively identify single neurons would be improved with larger format silicon probes. Our work employs compact tetrode electrodes that permit simultaneous resolution of single units and mesoscale GCAMP activity. It is conceivable that improvements in spike sorting fidelity could be made by switching to more densely spaced silicon probes. While this is an obvious advantage, these probes do not have a compact footprint and would interfere with regional imaging.

1. The authors present little analysis justifying the advantage of conducting chronic electrophysiological recordings instead of acute recordings with their data. In fact, throughout the paper, the authors mention that the results were consistent with their previous work with acute recordings. The only longitudinal analysis in this paper is qualitative and suggests that cortical maps were stable over days. I believe this was also shown in the past already. More in depth analysis of across days dynamics or showcase of an experiment centered on across days dynamics will strengthen the appeal of this approach. Generally speaking, there is very little quantitative analysis of longitudinal maps/functional coupling of single neurons over days. The paper will benefit from at least some quantification of this part.

To our knowledge data showing the persistence of spike-associated maps longer than an acute experiment is novel. However, due to a low yield of recorded single neurons, we have not been able to follow these maps over a longer period in a population that would permit group statistics. We suggest that future experiments could be done using silicon probes with larger yields which would help to better align electrophysiological features with mesoscale GCAMP maps.

1. Recording with tetrodes gives very low yields compared to silicon probe recordings. While silicon probes have a larger footprint and may occlude the widefield imaging on the side of the silicon probe implant, it is unclear why not to use denser electrode arrays on one side of the brain and image from the other hemispheres, given that the maps are very correlated across hemispheres

Taking advantage of mirrored activity in the opposite hemisphere is a great idea. Future studies could include experiments that would take advantage of bilateral symmetry by placing high-resolution silicon probes in one hemisphere and then reading out mesoscale maps in the other.

1. The advantage of the electrophysiological recordings is in providing access to single-neuron activity at high temporal resolution. The authors could add more quantifications regarding individual neuron functional coupling diversity. For instance, in the per-area distributions in Figure 5D -- did all neurons from a given area participate in the same functional maps, or did different neurons show diversity in the functional coupling. Did simultaneous recordings of neurons from the same tetrode show more similar maps, than recordings of other neurons from the same area conducted on different days/in different animals? Did the map differ when the neurons were bursting/were at specific phases of the LFP, etc.

Unfortunately the yield of neurons was not enough to investigate some of the interesting state-dependent phenomena the reviewer describes. In previous work we have examined heterogeneity between single neuron responses in more detail Xiao et al. 2027 in acute work.

1. Facial nerve stimulation. This part feels detached from the rest of the paper and is not explained/discussed in sufficient detail. For example, there is no description of the surgical procedure or the electrode used for facial nerve recordings in the Methods (in the Results section, the authors mention 'micro-wires', but the Method section only contains information about tetrodes).

Thank you for bringing up the issue of surgical details for facial nerve experiments are now in the methods. This information is also available by contacting the authors and below.

For facial nerve recordings, peripheral nerve activity was measured by fine wire recording directly from the nerves subserving the whisker. During surgery, mice will be anesthetized and positioned on a warming pad connected to a rectal probe, and the temperature maintained at 37 °C. A skin incision was made, exposing a small part of the buccal branch of the left facial nerve. Magnification of the surgical field with a dissecting microscope allowed a careful dissection of a nerve branch with minimum disruption of the tissues and blood supply surrounding the nerve. The appropriate site of exposure was determined by using two projection lines: a vertical line running downward, posterior from the outer corner of the eye, and a horizontal line running in the caudal direction, starting at the whisker E-row. Then two insulated fine wires (about 25 µm tips) were hooked and placed around the nerve separated about 2 mm from one another. The insulation at the ends of the wires was removed and a knot was made on each wire to prevent it from slipping. The opposite ends of each wire were soldered to a mini connector attached by dental cement to the skull. Finally, 6-0 silk sutures were used to close the skin incisions.

The functional maps associated with facial nerve spiking show different patterns from the optogenetic stimulation maps that led to significant facial nerve responses. Specifically, the STM maps show responses in the posterior parts of the cortex, but the photostimulation map showed almost an opposite pattern, where the effects were observed in the anterior parts. The authors do not discuss this mismatch in sufficient detail. Further, the authors refer to area PTA but use partitions based on the Allen Institute, which does not indicate this area.

The posterior parietal area location is based on our previous work Mohajerani et al. 2013 and using the Allen Institute Brain Atlas for guidance.

Minor comments1. The authors mention that "on average, we obtained 3-5 neurons per tetrode implanted, and this yield was consistent across regions (Figure 2C). " -- for how long, on average, could the authors record single-neuron activity from each tetrode?

The 3-5 neurons obtained per tetrode were recorded 1 week after tetrode implantation.

1. Figure 4B - it is unclear what the labels "recording 1, ...5, " correspond to. Are these different recording sessions within the same day "day 8"?

The labels "recording 1, ...5, " correspond to different recording sessions within the same day.